# Validation of the Emotiv EPOC® EEG gaming system for measuring research quality auditory ERPs

Nicholas A. Badcock[1], Petroula Mousikou[1,2], Yatin Mahajan[1], Peter de Lissa[1], Johnson Thie[3] and Genevieve McArthur[1]

[1] ARC Centre of Excellence in Cognition and its Disorders, Macquarie University, Sydney, NSW, Australia
[2] Department of Psychology, Royal Holloway, University of London, London, United Kingdom
[3] School of Electrical and Information Engineering, University of Sydney, Sydney, NSW, Australia

Corresponding author
Nicholas A. Badcock,
nicholas.badcock@mq.edu.au

## ABSTRACT

**Background.** Auditory event-related potentials (ERPs) have proved useful in investigating the role of auditory processing in cognitive disorders such as developmental dyslexia, specific language impairment (SLI), attention deficit hyperactivity disorder (ADHD), schizophrenia, and autism. However, laboratory recordings of auditory ERPs can be lengthy, uncomfortable, or threatening for some participants – particularly children. Recently, a commercial gaming electroencephalography (EEG) system has been developed that is portable, inexpensive, and easy to set up. In this study we tested if auditory ERPs measured using a gaming EEG system (Emotiv EPOC®, www.emotiv.com) were equivalent to those measured by a widely-used, laboratory-based, research EEG system (Neuroscan).

**Methods.** We simultaneously recorded EEGs with the research and gaming EEG systems, whilst presenting 21 adults with 566 standard (1000 Hz) and 100 deviant (1200 Hz) tones under passive (non-attended) and active (attended) conditions. The onset of each tone was marked in the EEGs using a parallel port pulse (Neuroscan) or a stimulus-generated electrical pulse injected into the O1 and O2 channels (Emotiv EPOC®). These markers were used to calculate research and gaming EEG system late auditory ERPs (P1, N1, P2, N2, and P3 peaks) and the mismatch negativity (MMN) in active and passive listening conditions for each participant.

**Results.** Analyses were restricted to frontal sites as these are most commonly reported in auditory ERP research. Intra-class correlations (ICCs) indicated that the morphology of the research and gaming EEG system late auditory ERP waveforms were similar across all participants, but that the research and gaming EEG system MMN waveforms were only similar for participants with non-noisy MMN waveforms ($N = 11$ out of 21). Peak amplitude and latency measures revealed no significant differences between the size or the timing of the auditory P1, N1, P2, N2, P3, and MMN peaks.

**Conclusions.** Our findings suggest that the gaming EEG system may prove a valid alternative to laboratory ERP systems for recording reliable late auditory ERPs (P1, N1, P2, N2, and the P3) over the frontal cortices. In the future, the gaming EEG

system may also prove useful for measuring less reliable ERPs, such as the MMN, if the reliability of such ERPs can be boosted to the same level as late auditory ERPs.

## INTRODUCTION

An auditory event-related potential (ERP) reflects the average electrical response of a large groups of brain cells in response to a particular sound (e.g., a high-pitched tone). This electrical activity can be measured at the scalp using electrodes. The first three positive peaks in an ERP waveform are often referred to as the P1, P2 and P3 (also referred to as the P100, P200, and P300), and the first two negative peaks are often called the N1 and N2 (N100 and N200, see Fig. 2, F3 panels). These "late auditory ERPs" are thought to be generated by neurons (i.e., brain cells) that process the physical features of sensory stimuli, and neurons involved in the detection, classification, and inhibition of stimuli (*Key, Dove & Maguire, 2005*). The auditory ERP waveform is therefore considered to reflect post-synaptic electrical activity predominantly in the primary and secondary auditory cortices (*Oades, Zerbin & Dittmann-Balcar, 1995*; *Tonnquist-Uhlén et al., 1996*).

One advantage of auditory ERPs is that it is possible to measure them passively, without a listener's attention; for example, whilst participants watch their favourite DVD. The undemanding nature of passive auditory ERPs has made them a popular tool for measuring auditory processing in inattentive listeners, such as children or adults with cognitive disorders such as developmental dyslexia (*McArthur, Atkinson & Ellis, 2009*), specific language impairment (*Barry et al., 2008*), autism (*McPartland et al., 2004*), attention-deficit hyperactivity disorder (*Taylor et al., 1997*), and schizophrenia (*Todd, Michie & Jablensky, 2003*). One ERP component commonly measured in these populations is mismatch negativity (MMN). This is calculated by subtracting a late auditory ERP to a rare "deviant" sound (e.g., high tone) from a late auditory ERP to a frequent "standard" sound (e.g., a low tone; see Fig. 3). This ERP is traditionally thought to reflect pre-attentive memory and auditory discrimination (*Näätänen, 1992*). However, recent research suggests that it may reflect N1 activity related to new afferent neuronal activity (*May & Tiitinen, 2010*).

A limitation of auditory ERPs is that they are typically measured in an experimental laboratory full of medical-looking equipment, which can be frightening for some people, such as children or adults with cognitive disorders. Further, it can take an experimenter 30–40 min to place 32 electrodes on a participant's scalp, making an ERP measurement session lengthy. Another limitation of ERPs is that many ERP electrode caps use a thick gel to create a connection between the scalp and each electrode. At the end of a session, a person is left with clumps of gel throughout their hair that can only be properly removed by thoroughly washing the entire head.

In recent times, the commercial computer gaming industry has come up with a tantalizing solution to these problems: wireless EEG systems (Emotiv EPOC®, Imec's wireless EEG headset, NeuroFocus Mynd™, Neurokeeper's headset, NeuroSky Mindwave®). These "gaming EEG systems" use EEG activity to control the movement of characters or objects in games via headsets that comprise a small array of sensors that (1) are wirelessly connected to software that runs on a laptop (so no need for an expensive laboratory); (2) require little adjustment of electrodes (so no need for long electrode placement procedures); and (3) typically use small cotton pads that are soaked in saline solution to connect each electrode to the scalp (so no need for messy gel, and hence no need for head washing).

A handful of researchers have tested the validity of using gaming EEG systems as research tools. Work by *Thie, Klistorner & Graham (2012)* and *Debener et al. (2012)* suggests that these systems are valid tools for measuring visual evoked potentials and EEG activity when walking outside a laboratory. However, to our knowledge, no study has validated the use of a gaming EEG system for measuring ERPs, perhaps because this requires a physical modification of the headset to insert stimulus markers into an EEG (see Equipment section below). Thus, the aim of the current study was to test the validity of one gaming EEG system as a measure of auditory ERPs, in an endeavor to minimize the stress associated with ERP measurement for some people, and allow the measurement of ERPs outside of laboratory settings (e.g., schools and audiologists' practices).

## MATERIALS AND METHODS

### Participants

The Macquarie University Human Research Ethics Committee approved the methods used to test participants (Ethics Ref: 5201200658). All participants gave written informed consent to be involved in the research. Twenty-one typical adults (12 female and 9 male) were involved in this study. The mean age of the participants was 31 years (SD = 5.3 years). Individuals had normal or corrected-to-normal vision and no history of auditory-related problems.

### Stimuli

Stimuli were delivered to participants binaurally at a comfortable listening volume through Phillips SHS4700/37 ear-clip headphones fixed to the gaming EEG system headset (see Fig. 1 for a diagram), using Presentation (version 16; Neurobehavioral System Inc.). There were two blocks of stimuli. Both blocks comprised 566 standard tones (175-ms 1000-Hz pure tones with a 10-ms rise and fall time; 85% of trials) and 100 deviant tones (175-ms 1200-Hz pure tones with a 10-ms rise and fall time; 15% of trials). After 3 standard tones (i.e., first 3 trials), the presentation of the deviant tones was randomly separated by 3–35 standard tones. The stimuli were separated by a jittered stimulus-onset synchrony (SOA) of 0.9–1.1 s to minimize the ERP related to the anticipation of a stimulus.

The first block of stimuli was presented in a "passive condition", in which participants were instructed to watch a silent video and ignore the tones coming through the ear-clips. The second block of stimuli was presented in an "active condition", in which participants

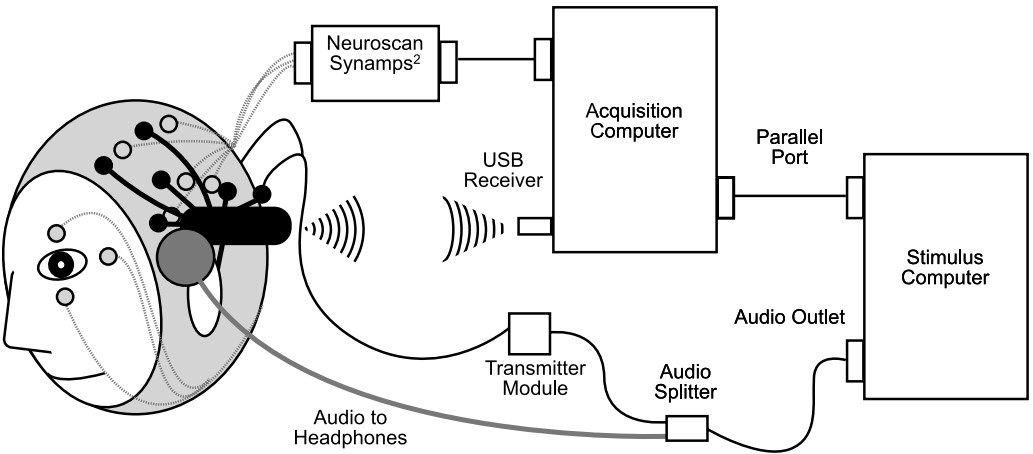

**Figure 1** Schematic diagram of simultaneous research EEG system (Neuroscan Synamps[2], in grey) and gaming EEG system (Emotiv EPOC, in black) setup.

were asked to count the deviant (higher) tones whilst watching the silent video. They were told that they would be asked to report the total number of deviant tones at the end of the session.

## Equipment

The Neuroscan system (version 4.3, hereafter referred to as the research EEG system) was connected to an EEG electrode cap that comprised 16 sintered Ag–AgCl electrodes that were sewn into a material cap according to the International 10–20 system (Easy Cap). We recorded EEGs from 16 sites: F3, F7, FC4, FT7, T7, P7, O1, O2, P8, T8, FT8, FC4, F8, F4, M1, and M2. The left mastoid (M1) served as online reference, which left us with 15 electrode sites (i.e., EEG channels). Vertical eye movements (VEOG) were measured with electrodes placed above and below the left eye. Horizontal eye movements (HEOG) were recorded using electrodes placed on the outer canthi of each eye. The ground electrode was positioned between FPz and Fz. The online EEG was sampled at 1000 Hz with an online bandpass filter from 1 to 100 Hz. During offline processing, the research EEG system data was downsampled (using EEGLAB; *Delorme & Makeig, 2004*) to 128 Hz to make it compatible with the gaming EEG system sample rate. The onset of each stimulus was marked in the EEG using parallel port pulses.

The gaming EEG system (i.e., Emotiv EPOC®) used gold-plated contact-sensors that were fixed to flexible plastic arms of a wireless headset (see Fig. 1). The headset included 16 sites, aligned with the 10–20 system: AF3, F7, F3, FC5, T7, P7, O1, O2, P8, T8, FC6, F4, F8, FC4, M1, and M2. One mastoid (M1) sensor acted as a ground reference point to which the voltage of all other sensors was compared. The other mastoid (M2) was a feed-forward reference that reduced external electrical interference. The signals from the other 14 scalp sites (channels) were high-pass filtered with a 0.16 Hz cut-off, pre-amplified and low-pass filtered at an 83 Hz cut-off. The analogue signals were then digitised at 2048 Hz. The digitised signal was filtered using a 5th-order sinc notch filter (50–60 Hz), low-pass filtered

and down-sampled to 128 Hz (specifications taken from the gaming EEG system web forum). The effective bandwidth was 0.16–43 Hz.

Because the gaming EEG system was developed for measuring EEG and not ERPs, it had to be modified to send markers to the EEG to indicate the onset of each stimulus. This was done using a custom-made transmitter module (see below for a description) that converted each tone presented by Presentation into a positive and negative electrical signal. These signals were injected into the O1 and O2 channels of the gaming EEG system headset. The resulting positive and negative spikes recorded in the O1 and O2 EEGs were processed offline in Matlab. A between-channels difference greater than 50 mV was coded as a stimulus onset or offset. The event marker was set at a constant time interval (5 ms delay of the transmitter module) prior to the point of positive and negative signal cross-over. For around 10% of subjects, there were more stimulus markers automatically identified than expected, which was due to participant movement. These markers were removed manually. The valid stimulus markers were recombined with the EEG data. The "sacrifice" of the O1 and O2 sites for the purposes of stimulus marking left us with 12 gaming EEG system scalp electrode sites.

The custom-made transmitter module consists of two parts: audio tone detector and pulse generator. The audio tone detector is connected to the headphone output of the Presentation computer. Upon detecting an audio tone, it activates the pulse generator. The pulse generator in turn generates a pulse to O1 while O2 is treated as the isolated ground. The amplitude of the pulse is adjustable from 30 to 300 μV. The pulse duration is fixed at 35 ms. The pulse generator is electrically isolated from the audio tone detector by using an opto-coupler. This ensures that the subject is electrically isolated from any equipment connected to the mains, which is the Presentation computer in this case.

## Testing procedure

Each participant was seated in a comfortable chair 1.5 m away from a 13-inch laptop screen. The experimenter combed the participants on the left, mid-line, and right sides of their scalp firmly in order to reduce electrode impedances (*Mahajan & McArthur, 2010*). After the relevant areas on the face and mastoids had been cleaned, the Easy Cap (connected to the research EEG system) was positioned on the participant's head. A blunt metal tube mounted in a syringe was then used to inject conductive gel into the small gap that separated each electrode from the participant's scalp. The blunt tube was circled and gently rocked back and forth against the scalp 3 to 4 times before injecting the gel. This took around 30–45 min depending on the participant's scalp conductivity.

Once all scalp electrodes had been filled with gel, the impedances of each electrode were measured using Neuroscan Synamps[2] acquisition system and the Scan software (Scan 4.3). Further adjustments were made to each electrode until the impedance for all electrodes was less than 5 kΩ. The gaming EEG system headset was then fitted on top of Easy Cap. The gaming EEG system sensors were placed on the head through custom slits made in the Easy Cap (see Fig. 1). The custom slits were made by fitting the gaming EEG system headset over the Easy Cap and making incisions where the gaming EEG system sensors fell. Where the

Table 1 **Median number of accepted epochs for research and gaming EEG systems by condition and tone type.** Median (inter-quartile range) trial numbers for the research and gaming EEG systems in each Condition (Passive versus Active listening) and for each Tone type (Standard, Deviant, and Total). Wilcoxon Signed Rank Test $Z$ values are also presented.

| | | EEG System | | |
| Condition | Tone | Research | Gaming | $Z$ |
| --- | --- | --- | --- | --- |
| Passive | Standard | 564 (3) | 558 (18) | 3.7[*] |
| | Deviant | 100 (1) | 98 (1) | 3.11[*] |
| | Total | 663 (3) | 657 (21) | |
| Active | Standard | 563 (5) | 559 (11) | 3.33[*] |
| | Deviant | 100 (1) | 98 (3) | 2.15 |
| | Total | 663 (7) | 658 (10) | |

**Notes.**

[*] $p < 0.0125$ Bonferonni corrected for 4 comparisons.

system positions overlapped, slits were made directly adjacent to the Easy Cap electrode location. The sensors were adjusted until suitable connectivity was achieved as indicated by the TestBench software, which adds a small modulation to the feedforward signal, and measures the size of the signal back from each channel. This procedure took 10–15 min.

The research and gaming systems recorded EEG simultaneously, as opposed to recording with one system and then the other in a counterbalanced fashion, to maximise the conditions for validation. Specifically, if separate recordings were made and differences were noted, it would be difficult to determine if these were due to differences between the systems, or due to differences between the state of the brain at different points in time (e.g., differences in level of fatigue, or differences in the amount of exposure to the stimuli).

## Offline EEG processing

The research and gaming system EEGs were processed offline using EEGLAB version 11.0.4.3b (*Delorme & Makeig, 2004*). Major artifacts were first excluded by eye, and then the EEG in each channel was bandpass filtered from 0.1 to 30 Hz. Eye-movements and heartbeat signals (heartbeat signals were present in the research system EEG for just 5 subjects) were removed using independent components analysis (ICA) in EEGLAB ('eeg_runica' function). The cleaned EEG signals in each channel were then cut into epochs that started −102 ms before the onset of each stimulus (0 ms), and ended 500 ms after the onset of the same stimulus. Each epoch was baseline corrected from −102 to 0 ms. Epochs with amplitude values ±150 μV were excluded. The median number of accepted epochs in the research and gaming EEG system waveforms is shown in Table 1.

## Creating waveforms

Accepted epochs were averaged together to create a standard late ERP waveform and a deviant late ERP waveform (see Fig. 2) for both active and passive conditions at each scalp site for each participant. The standard late ERP waveforms in both active and passive conditions were used to measure the P1, N1, P2, and N2 peaks (deviant late ERP waveforms were not used because these comprised fewer epochs, and hence were

less reliable than standard ERP waveforms). The deviant late ERP waveform in the active condition was used to represent the P1, N1, P2, and N2 peaks as well as the P3 ERP peak since the P3 is enhanced by active attention to a rare or unexpected event. Note that we define ERP peaks and components in line with *Luck (2005)*. That is, P1, N1, P2, N2, and P3 ERPs are peaks, and the MMN is a component.

The late auditory ERP waveforms in the passive condition were used to create one MMN waveform per person. We did this by subtracting the standard late auditory ERP waveform from the deviant late auditory ERP waveform (see Fig. 3). Data from the passive condition was used because directing attention away from the experimental stimuli (i.e., towards a movie) minimises the impact of confounding attention-related P3a and P3b ERPs on the MMN (*Lang et al., 1995*; *Sinkkonen & Tervaniemi, 2000*).

### Measuring waveform peaks

The P1, N1, P2, and N2 ERP peaks were represented by the first two clear positive peaks (P1 and P2), and the first two clear negative peaks (N1 and N2) in the passive and active standard late auditory ERP waveforms (see Fig. 2). For all subjects, P1 fell between 15 and 120 ms; N1 fell between 60 and 180 ms; P2 fell between 100 and 420 ms; and N2 fell between 170 and 490 ms. The amplitude and latency of each peak for each participant was measured via manual selection of the peaks using the EEGLAB software. Manual selection was used for all peaks because automated peak classification incorrectly classified peaks in a minority of individuals due to variation in individuals' waveforms. For example, the waveforms of a few participants had a noisy baseline that resulted in incorrect detection of P1. In this early stage of research on gaming EEG systems, we felt it wise to err on the side of caution, and thus manually measured peaks for each participant individually to ensure that all latency and amplitude estimates were truly valid.

The P3 ERP was represented by the third positive peak in the deviant late auditory ERP waveform in the active condition (see Fig. 2). The amplitude and latency of the P3 peak for each participant was measured via manual selection of the peak using the EEGLAB software. For all subjects, this interval fell between 210 and 420 ms.

The MMN was represented by the large negative deflection in the passive MMN waveform. It was estimated as the minimum voltage (i.e., a negative peak) in the interval across which the MMN waveform fell below 0 (i.e., when the MMN was present). For all subjects, this interval fell between 80 and 250 ms. However, it is noteworthy that the research and gaming EEG system MMN could not be calculated for two participants, and the gaming EEG system MMN could not be calculated for an additional eight participants. In these cases, the deviant late ERP waveform was unexpectedly (and in some cases, drastically) more positive than the standard late auditory ERP waveform.

## RESULTS

For ease of presentation and hence understanding, all analyses focussed on data from two frontal sites in the left and right hemispheres: F3 and F4 for the research EEG system, AF3 and AF4 for gaming EEG system. We chose these sites because they typically register the largest late auditory ERP and MMN responses, and hence are most commonly used to

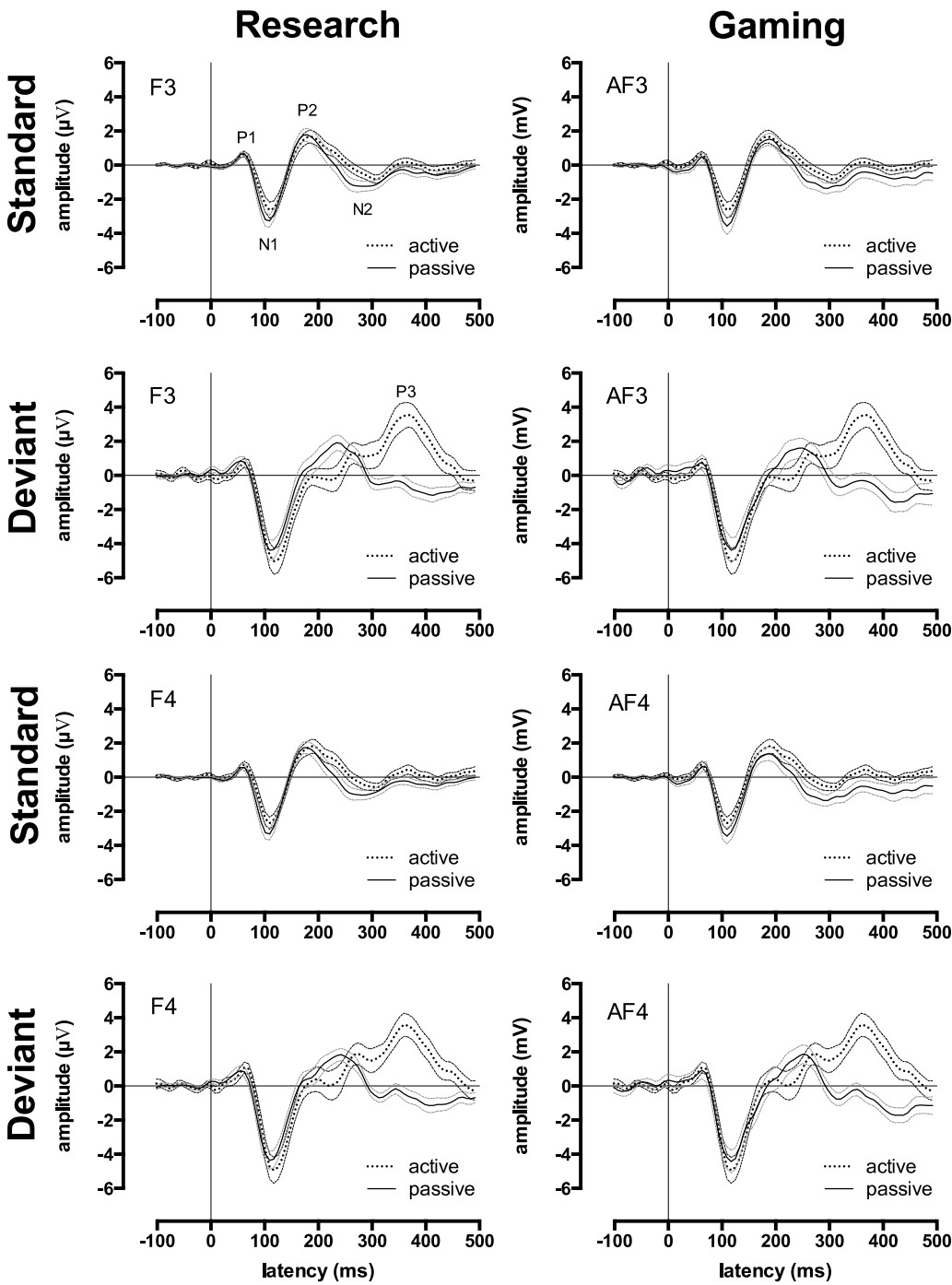

**Figure 2 Research and gaming system ERP waveforms by condition, tone type, and hemisphere.**
Group ERP waveforms for research (left-side) and gaming (right-side) systems. All graphs display wave-forms for the passive and active (counting deviant tones) listening conditions. The upper 4 graphs depict the left-hemisphere-activity (F3 and AF3) and the lower 4 graphs depict the right-hemisphere-activity (F4 and AF4). Rows 1 and 3 depict waveforms elicited by the standard tones, rows 2 and 4 depicts waveforms elicited by the deviant tones. Error waveforms (in grey) represent the standard error of the mean.

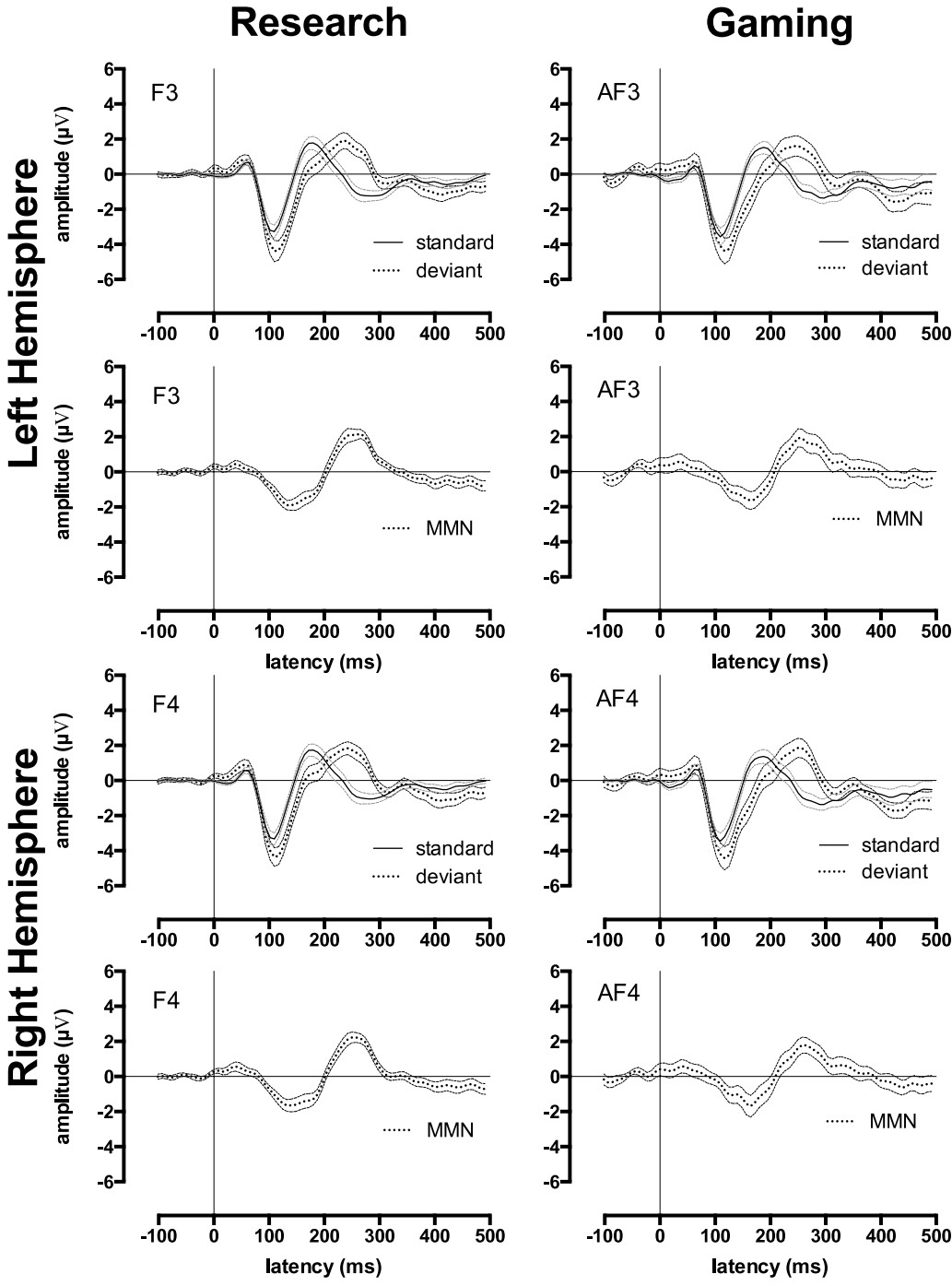

**Figure 3 Research and gaming system Mismatch Negativity related waveforms by hemisphere.** Group ERP and Mismatch Negativity (MMN) waveforms for research (left-side) and gaming (right-side) systems. All graphs display waveforms for the passive listening condition. The upper 4 graphs depict the left-hemisphere-activity (F3 and AF3) and the lower 4 graphs depict the right-hemisphere-activity (F4 and AF4). Rows 1 and 3 depict waveforms elicited by the standard tones and deviant tones, rows 2 and 4 depict MMN waveforms (deviant minus standard waveforms). Error waveforms (in grey) represent the standard error of the mean.

**Table 2 Research versus gaming EEG system ERP and MMN waveform Intraclass Correlations.** Mean intraclass correlations (ICC) and 95% confidence intervals between waveforms simultaneously recorded with the research and gaming EEG systems for the left (F3/AF3) and right (F4/AF4) hemispheres. ICCs are presented for the passive and active listening conditions as well as the standard and deviant tones. For the passive condition, the ICCs for the deviant minus standard waveforms, the mismatch negativity (MMN), is also presented ($n = 21$ but see note a).

| Condition | ERP | Hemisphere | |
| | | F3/AF3 | F4/AF4 |
| --- | --- | --- | --- |
| Passive | Standard | 0.74 (0.12) | 0.74 (0.11) |
| | Deviant | 0.57 (0.18) | 0.67 (0.14) |
| | MMN | 0.44 (0.17) | 0.44 (0.19) |
| | MMN[a] | 0.71 (0.16) | 0.71 (0.19) |
| Active | Standard | 0.79 (0.12) | 0.8 (0.09) |
| | Deviant | 0.77 (0.08) | 0.8 (0.08) |

Notes.

[a] $n = 11$, exclusion based on manual evaluation of waveform reliability (i.e., spikes of noise rather than smooth waveform).

represent these ERPs in the research literature (*Bishop et al., 2007*; *Ponton et al., 2000*). The mean research and gaming EEG system late auditory ERP waveforms for standard and deviant stimuli in the passive and active conditions at these sites are shown in Fig. 2. Morphologically, these waveforms were all consistent with mature auditory ERPs (*Bishop et al., 2007*; *Mahajan & McArthur, 2012*; *Ponton et al., 2000*).

## Waveform reliability

We tested the reliability of the waveforms produced by the research and gaming EEG systems using the number of epochs accepted into the standard and deviant waveforms in the passive and active conditions. These are shown in Table 2.

Due to significant negative skews in the data, we compared the accepted number of epochs for each system using Wilcoxon Signed Rank Tests and a Bonferroni corrected $p$-value for multiple comparisons ($p = .0125$). There were significantly fewer epochs accepted for all gaming EEG system waveforms except for the deviant waveform in the active condition. However, the median values in Table 1 show that there was, in fact, very little difference in the median number of epochs accepted for the research and gaming EEG systems. Further, the number of accepted epochs for both systems was more than adequate. Thus, the gaming EEG system produced reliable waveforms - even if slightly less reliable than the research EEG system.

## Waveform similarity

We tested the similarity of the research and gaming EEG system waveforms using intraclass correlations (ICC) in line with *Bishop et al. (2007)*. ICCs reflect the similarity of waveforms in terms of morphology, amplitude, and latency. We used the ICCs to measure similarity across the entire waveform (i.e., $-102$ ms to 500 ms). The ICC values were considered statistically significant if 95% confidence intervals did not include 0.

Table 2 shows the ICCs between the research and gaming EEG system late auditory ERP waveforms in active and passive conditions. For the ERP waveforms to the standard tones, the ICCs for the active and passive waveforms ranged from 0.74 to 0.80 (depending on scalp site and active or passive conditions), indicating a strong similarity between the research and gaming EEG system recordings. This was supported by the 95% confidence intervals, which indicated that these ICCs were statistically significant.

For the ERP waveforms to the deviant tones in the active condition (including the P3 peak), the ICCs were 0.77 and 0.80 for left and right frontal sites, respectively. These high ICCs were statistically significant. Thus, the ERP waveforms to the deviant tones in the active condition measured by the research and gaming EEG systems were similar in morphology.

For the MMN waveform, presented in Fig. 3, which was measured in the passive condition, ICC outcomes were calculated twice. The first calculation, which included all participants, indicated that the ICCs between research and gaming EEG system MMN waveforms were statistically significant (ICC = 0.44). However, these MMN waveforms were markedly less similar than the ERP waveforms to the standard and deviant tones. The second calculation, which excluded participants with noisy MMN waveforms (leaving $N = 11$), revealed a stronger mean ICC between the two systems (ICC = 0.71). Thus, the MMN waveforms measured by the two systems were similar for participants who produced reliable MMN waveforms. However, reliable waveforms were not produced by half of the adults in this sample.

## Waveform peaks

The research and gaming EEG system P1, N1, P2, N2, P3, and MMN peak amplitude and latency measures did not differ from a normal distribution and so were compared using paired sample $t$-tests. Descriptive and inferential statistics for the left and right hemispheres are presented in Table 3. All bar one comparison indicated no statistical difference between the mean peak amplitudes and latencies for the research and gaming EEG system P1, N1, P2, and N2 ERPs. The exception was the P2 latency in the active condition, which was later for gaming EEG system. Further, Cohen's $d$ measures that indexed the size of the difference between the research and gaming EEG system measures were typically small for both amplitude and latency in the left (0.18 and 0.44 respectively) and right (0.23 and 0.47 respectively) frontal sites. Interestingly, the Cohen's $d$ measures also revealed slightly smaller amplitude differences in passive than active conditions (passive standard = .20 and deviant = .14; active standard = .26 and deviant = .21).

In addition, condition by stimulus comparisons for the research and gaming systems showed no significant differences. The descriptive and inferential statistics for these comparisons are included in Tables S1 and S2: 1 for the left hemisphere comparisons, and 2 for the right.

The descriptive and inferential statistics for the P3 are presented in Table 4. The research and gaming EEG system P3 amplitudes and latencies were not statistically different, and

**Table 3 Research versus gaming EEG system ERP peak comparisons.** Descriptive (M and SD) and inferential ($t$ and Cohen's $d$) statistics for peak (P1, N1, P2, N2) amplitude (μV) and latency (ms) for research versus gaming EEG system comparisons in the passive and active listening conditions across both hemispheres, denoted by site ($n = 21$).

| Condition | ERP | Measure | Site | EEG system | | $t$ | $d$ |
|---|---|---|---|---|---|---|---|
| | | | | Research | Gaming | | |
| Passive | P1 | Amplitude | F3/AF3 | 0.76 (0.72) | 0.68 (1.01) | −0.36 | 0.10 |
| | | | F4/AF4 | 0.75 (0.79) | 0.53 (1.42) | −0.74 | 0.19 |
| | | Latency | F3/AF3 | 58.6 (12.35) | 60.62 (22.53) | 0.41 | 0.11 |
| | | | F4/AF4 | 59.71 (14.6) | 63.88 (20.17) | 1.42 | 0.24 |
| | N1 | Amplitude | F3/AF3 | −3.14 (1.59) | −2.77 (2.18) | 0.91 | 0.2 |
| | | | F4/AF4 | −3.36 (1.71) | −2.79 (2.34) | 1.18 | 0.29 |
| | | Latency | F3/AF3 | 104.86 (8.1) | 110.85 (17.85) | 1.74 | 0.45 |
| | | | F4/AF4 | 106.29 (8.31) | 111.96 (18) | 1.71 | 0.42 |
| | P2 | Amplitude | F3/AF3 | 1.85 (1.33) | 1.36 (1.31) | −1.55 | 0.39 |
| | | | F4/AF4 | 1.87 (1.42) | 1.47 (1.53) | −1.16 | 0.28 |
| | | Latency | F3/AF3 | 177.3 (14.9) | 196.39 (27.98) | 3.45 | 0.88 |
| | | | F4/AF4 | 176.78 (15.69) | 196.91 (35.26) | 3.21 | 0.76 |
| | N2 | Amplitude | F3/AF3 | −1.37 (1.4) | −1.3 (1.61) | 0.2 | 0.05 |
| | | | F4/AF4 | −1.34 (1.48) | −1.24 (1.4) | 0.33 | 0.07 |
| | | Latency | F3/AF3 | 277.95 (24.59) | 280.17 (37.08) | 0.5 | 0.07 |
| | | | F4/AF4 | 278.15 (28.16) | 284.73 (41.05) | 0.88 | 0.19 |
| Active | P1 | Amplitude | F3/AF3 | 0.87 (0.67) | 0.71 (0.97) | −0.87 | 0.2 |
| | | | F4/AF4 | 0.99 (0.62) | 0.72 (0.88) | −1.83 | 0.36 |
| | | Latency | F3/AF3 | 59.64 (15.18) | 63.81 (20.5) | 0.83 | 0.24 |
| | | | F4/AF4 | 57.23 (19.94) | 63.16 (18.72) | 1.84 | 0.32 |
| | N1 | Amplitude | F3/AF3 | −2.62 (1.58) | −2.26 (1.94) | 1.01 | 0.21 |
| | | | F4/AF4 | −2.78 (1.55) | −2.18 (1.69) | 1.75 | 0.39 |
| | | Latency | F3/AF3 | 107.72 (15.84) | 115.41 (17.73) | 1.68 | 0.47 |
| | | | F4/AF4 | 108.31 (18.27) | 115.8 (18.35) | 2.15 | 0.42 |
| | P2 | Amplitude | F3/AF3 | 2.12 (1.39) | 1.84 (1.43) | −1.1 | 0.21 |
| | | | F4/AF4 | 2.35 (1.52) | 1.7 (1.51) | −3.23 | 0.45 |
| | | Latency | F3/AF3 | 180.17 (17.27) | 198.47 (25.8) | 3.87[*] | 0.86 |
| | | | F4/AF4 | 183.1 (20.78) | 201.4 (29.7) | 3.57 | 0.74 |
| | N2 | Amplitude | F3/AF3 | −0.85 (1.18) | −0.66 (1.11) | 0.96 | 0.17 |
| | | | F4/AF4 | −0.71 (1.13) | −0.64 (0.93) | 0.42 | 0.07 |
| | | Latency | F3/AF3 | 283.23 (26.61) | 290.65 (27.56) | 1.8 | 0.28 |
| | | | F4/AF4 | 273.52 (32.95) | 292.8 (27.41) | 2.85 | 0.66 |

**Notes.**
[*] $p < .0015$, Boneferonni corrected for 32 comparisons.

the differences were small (all Cohen's $d < 0.24$) for peak and latency measures at left and right frontal sites.

The descriptive and inferential statistics for the MMN are presented in Table 4. Across all participants, the difference between the mean amplitude for the research and gaming EEG system MMNs at left and right frontal electrodes was not statistically significant.

**Table 4  Research versus gaming EEG system P3 and MMN comparisons.** Descriptive (M and SD) and inferential ($t$, Cohen's $d$) statistics for P3 peak amplitude ($\mu$V) and latency (ms) and Mismatch Negativity (MMN) amplitude ($\mu$V) for research (F3/F4) versus gaming (AF3/AF4) EEG system comparisons ($n = 21$ but see notes a and b).

| ERP | Measure | Site | EEG System | | $t$ | $d$ |
|---|---|---|---|---|---|---|
| | | | Research | Gaming | | |
| P3 | Amplitude | F3/AF3 | 3.61 (3.1) | 4.32 (3.39) | 1.13 | 0.23 |
| | | F4/AF4 | 3.48 (3.13) | 3.56 (2.81) | 0.17 | 0.03 |
| | Latency | F3/AF3 | 333.39 (52.68) | 327.14 (55.23) | −0.85 | 0.12 |
| | | F4/AF4 | 328.96 (49.68) | 326.1 (56.01) | −0.46 | 0.06 |
| MMN[a] | Amplitude | F3/AF3 | −3.24 (1.92) | −3.17 (1.71) | −0.22 | 0.04 |
| | | F4/AF4 | −3.26 (2.34) | −2.82 (1.68) | −0.97 | 0.23 |
| | Latency | F3/AF3 | 159.8 (24.55) | 149.15 (26.48) | 1.97 | 0.45 |
| | | F4/AF4 | 153.41 (24.02) | 144.18 (19.88) | 2.08 | 0.45 |
| MMN[b] | Amplitude | F3/AF3 | −3.22 (2.25) | −2.94 (1.63) | −0.70 | 0.15 |
| | | F4/AF4 | −3.38 (2.53) | −2.69 (1.55) | −1.48 | 0.35 |
| | Latency | F3/AF3 | 163.02 (37.8) | 145.31 (23.77) | 1.78 | 0.59 |
| | | F4/AF4 | 150.52 (40.21) | 136.46 (24.86) | 1.71 | 0.44 |

**Notes.**

[a] $n = 15$, exclusion based on missing values (i.e., incalculable due to the deviant waveform being higher than standard).

[b] $n = 11$, exclusion based on manual evaluation of waveform reliability (i.e., spikes of noise rather than smooth waveform).

However, the difference between the research and gaming EEG system MMN amplitude was noticeably larger (Cohen's $d = 0.37$) than for other ERPs (Cohen's $d$ 0.18 to 0.23). Removing 10 participants with noisy MMN waveforms from the analysis reduced this difference to Cohen's $d = 0.25$. Thus, the amplitude of the research and gaming EEG system MMN components was similar for participants whose MMN waveforms appeared to be reliable (i.e., uncontaminated by noise).

## DISCUSSION

The aim of the current study was to test the validity of a gaming EEG system as an auditory ERP research tool. To this end, we modified the gaming EEG headset so that it could insert stimulus markers into the EEG. We then took simultaneous EEG measurements using a research EEG system and a gaming EEG system in 21 adults who were presented with 666 standard and deviant tones under passive and active listening conditions.

The analyses were restricted to the frontal sites as these register the largest late auditory ERP responses and are most typically reported in the literature (*Bishop et al., 2007*; *Ponton et al., 2000*). We processed the EEG data offline to calculate the auditory ERP waveform to the standard tones, the ERP waveform to the deviant tones in the active condition, and the MMN waveform. In line with previous research, we used left and right frontal sites to represent these auditory ERPs. ICCs indicated that the ERP waveforms to both the standard and deviant tones were similar for the research and gaming EEG systems. This was not the case for the MMN waveform for a number of participants in the study.

A potential explanation for this finding is the lower reliability of the MMN compared to late auditory ERP (*Mahajan & McArthur, 2011*; *McArthur, Bishop & Proudfoot, 2003*): This lower reliability is likely due to the reduced number of epochs used to generate the deviant waveform, against which the standard waveform is subtracted, thus introducing error variance into the MMN. This explanation was confirmed by the removal of participants with particularly noisy MMN responses, which improved the ICC for the MMN waveform markedly. Thus, research and gaming EEG system MMN waveforms were only similar for participants with reliable MMN waveforms.

The ICCs also revealed that the deviant waveforms were less reliable in the passive ($M = .63$) than the active ($M = .79$) condition (see Table 2). A speculative explanation for this difference is movement interference. The passive condition was always completed first. With time, the gel of the research EEG system dries, providing a more stable bridge from the electrode to the scalp. This stability would be greatest in the active condition, completed second. Therefore, subtle participant movements, not necessarily affecting the number of accepted epochs (see Table 1) would have a greater influence in the passive condition. Because these movements are unlikely to affect the two systems equivalently, the result would be a reduced similarity between the waveforms measured with each system. This would be most evident to the deviant over the standard stimuli due to the deviant waveform being derived from fewer epochs.

The peak amplitude and latency measures supported the ICC outcomes. In all bar one case (P2 latency was later for the gaming EEG system in the active condition), there was no statistically significant difference between the amplitude and latency of the P1, N1, P2, N2, P3, and MMN peaks. However, there was a larger difference between the research and gaming EEG system MMN amplitude measures than the late auditory ERP measures. Further, the (non-significant) differences between the research and gaming EEG system late auditory ERP peak measures were smaller for the passive than active condition.

## CONCLUSIONS

Considered together, the results of this study suggest that the gaming EEG system compares well with the research EEG system for reliable auditory ERPs such as the P1, N1, P2, N2, and P3 measured at the frontal sites. In the future, Emotiv may also prove useful for measuring less reliable ERPs, such as the MMN, if the reliability of such ERPs can be boosted to the same level as late auditory ERPs. The apparent validity of the gaming EEG system for measuring reliable auditory ERPs, paired with its quick and clean set-up procedure and its portability, makes it a promising tool for measuring auditory processing in people from special populations who are unable or unwilling to be tested in an experimental laboratory. It may also open up new avenues for research since, in principle, the gaming EEG system can be used anywhere (e.g., schools, homes, shopping centres, hospitals) to measure brain responses in children and adults.

## ACKNOWLEDGEMENTS

We would like to thank participants who volunteered their time.

# PeerJ

### Funding

This research was supported by an ARC Centre of Excellence Grant [CE110001021] and an NHMRC equipment grant. The funders had no role in study design, data collection and analysis, decision to publish, or preparation of the manuscript.

### Grant Disclosures

The following grant information was disclosed by the authors:
ARC Centre of Excellence Grant: CE110001021.
NHMRC equipment grant: Internal Macquarie University scheme 2010.

### Competing Interests

Genevieve McAthur is an Academic Editor for PeerJ. None of the remaining authors have competing interests associated with the publication of this research.

### Author Contributions

- Nicholas A. Badcock and Petroula Mousikou conceived and designed the experiments, performed the experiments, analyzed the data, and wrote the paper.
- Yatin Mahajan conceived and designed the experiments and analyzed the data.
- Peter de Lissa and Genevieve McArthur conceived and designed the experiments, analyzed the data, and wrote the paper.
- Johnson Thie wrote the paper and engineered a critical piece of equipment (the transmitter module).

### Human Ethics

The following information was supplied relating to ethical approvals (i.e. approving body and any reference numbers):

The Macquarie University Human Research Ethics Committee approved the methods used to test participants [Ethics Ref: 5201200658].

### Supplemental Information

Supplemental information for this article can be found online at http://dx.doi.org/10.7717/peerj.38.

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
