# Peer review of "Validation of the Emotiv EPOC® EEG gaming system for measuring research quality auditory ERPs"

_PeerJ, doi:10.7717/peerj.38_

## Round 0.1 · original submission · Major Revisions

Please enclosed are comments to improve your manuscript which will be rereviewed after resubmission.

·

Basic reporting

The abstract contains the following “but that the Emotiv Emotiv and..”.

I do not believe that it is appropriate that the wireless device be referred to by the manufacturers name. Rather it should be identified as a product of the manufacturer (Emotiv) in the Method and referred to as “the wireless device (WD)” or similar throughout the MS. Similarly, the Neuroscan system should be identified in the Method and referred to as the “research system”.

Line 34 - the reference to Figure 1 should be removed as the figure shows the setup specific to this study only.

Line 37 - “generated neurons (i.e., by brain cells)” should be “generated by neurons (i.e., brain cells)”

Line 39 to 44 – The very brief introduction to ERP components only focuses on MMN, and does not refer to the most common recording situation for MMN (i.e. passive) until later (line 45) where it wrongly implies that all auditory ERPs are recorded in passively. This section should be clarified. I do not understand why the functional significance only MMN only is included.

Line 50 – ADD with and without hyperactivity is not a current diagnostic grouping.

Line 60 – there are several commercial devices available, with various combinations of wet/dry sensors, wired/wireless and numbers of electrodes. Please expand this section to present a broader view of the area and available tools.

Experimental design

Line 177 – What is a P1-N-P2-N2 waveform? What does the N refer to? Why not just refer to the ERPs using conventional labeling? In this case “the ERP to standard stimuli”. Similarly, in line 180 “the ERP deviant stimuli in the active condition”. All instances of this should be changed throughout the MS.

Line 177 – please use the term “components” to refer to the peaks contained within an ERP waveform, as per convention. For example, “the P3 ERP” should be “the P3 component”.

Line 191 to 193 - What are the ranges provided for ERP peak latency based on (observation, statistics)? Was a latency range determined from grand mean waveforms and used to guide peak selection?

Line 193 – Manual selection of peaks is open to selection biases. What is the justification for taking this approach over an automatic process based on latency search ranges?

Line 198 to 200 – the method used to quantify the MMN is entirely unclear, and must be improved and clarified.

Line 243 – please clarify the meaning of “…even though the ICC calculations were based on a waveform created from only 100 epochs in the deviant active condition.”

Validity of the findings

Analyses are restricted to frontal sites, and thus the general conclusion about the reliability of the so-called “Emotiv ERPs” is not supported, and should be altered. Alternatively, data could be presented from other scalp locations to present a more convincing view of reliability topographically.

Additional comments

Results - What about Stimulus effects (i.e. comparisons of deviant versus standard ERP component amplitude and latencies)? How reliable are they between the systems, in the active and passive tasks separately? The authors do not provide any indication of the reliability between systems of the comparison of ERP components in the deviant and standards (where possible). This would be a valuable addition to the paper.

Page 291 – perhaps the authors could speculate on possible causes of the lower reliability of MMN compared to auditory ERPs?

Table 2 caption mentions “… and action listening conditions…”

Table 3 shows quite poor ICC outcomes for the deviant ERP under passive conditions. This should be a topic for consideration in the Discussion.

Reviewer 2 ·

Basic reporting

In this study the authors observed the reliability and similarity in the different component of the late Auditory Event related potentials measured by two systems, one is standard laboratory system Neuroscan (version 4.3) and another one is Emotiv gaming system. The advantage of Emotiv system is in the application, it does not need the gel for EEG recording electrodes and thereby the subjects do not need to wash the head, on the contrary the Neuroscan system has these kinds of embarrassing experiences. The Emotiv EEG set up also take shorter time than Neuroscan. The paper is well written and well organized and easily understandable. The references are appropriate and contemporary.

Experimental design

The submission is in the scope of the journal. The aim of the research is clear, However, I am wondering about some minor issues as follows;

The authors recorded EEG from 15 sites out of 16 (since M1 was used for reference) by Neuroscan according to the international 10-20 system and from 12 sites out of 16 (since M1 & M2 for references and O1 & O2 for event marker) by Emotiv system. Though the investigator makes slit for Emotiv sensors on the Easy Cap, does it follow the international 10-20 system since the EEG recording was concurrent?

Validity of the findings

The findings of this paper are controlled but still some issues might needed to discuss as follows;

1) In the Introduction, line 39-40, the authors mentioned the rare deviant sound as low tone in bracket and standard sound as high tone; usually it happens in opposite way which is also observed in this study.

2) In the section of Equipment, at line 121, 128Hz1?

4) In the section of Creating Waveforms, in the line 177 and 180, P1-N-P2-N2, 1 is missing besides N.

3) In the Table 1, the total trial numbers are short from the actual number (e.g. 564+100, 558+98, 559+98), it is not clear.

5) Why not the authors recorded the EEG/ERP separately by these two systems and validate? This issue needs to discuss a little in this paper.

---

## Round 0.2 · Minor Revisions

Very Important to do :Original point 14. This point has not been addressed adequately. As I originally stated, your analyses are restricted to frontal sites, and thus the general conclusions about the reliability of the so-called “Emotiv ERPs” are not supported. The conclusions (and abstract reflection of them) must be altered to be explicitly specific to frontal sites only.

Reviewer 1 ·

Basic reporting

No Comments

Experimental design

No Comments

Validity of the findings

No comments

·

Basic reporting

The authors have addressed most of my listed concerns. Some remain however.

Original point 2. I do not agree that using the manufacturers name 8 times in the revised document aids in communication. The manufacturers name should be mentioned once in the Methods section (as it is at line 199 of the revised document).

Original point 7. You need to be fair and consistent when listing the other portable wireless EEG devices - mention the manufacturers name and device name for each, as you have with the emotiv device (e.g. NeuroSky Mindwave).

Experimental design

None.

Validity of the findings

Original point 14. This point has not been addressed adequately. As I originally stated, your analyses are restricted to frontal sites, and thus the general conclusions about the reliability of the so-called “Emotiv ERPs” are not supported. The conclusions (and abstract reflection of them) must be altered to be explicitly specific to frontal sites only.

---

## Round 0.3 · accepted · Accept

Thank You for your revised manuscript.We wish you well and that future scientific papers will be submitted to PeerJ.

·

Basic reporting

My 2nd round of concerns have been addressed.

Experimental design

none

Validity of the findings

none

Additional comments

none